# Remote sensing of methane leakage from natural gas and petroleum systems revisited

Oliver Schneising, Michael Buchwitz, Maximilian Reuter, Steffen Vanselow, Heinrich Bovensmann, and John P. Burrows

Institute of Environmental Physics (IUP), University of Bremen FB1, Bremen, Germany

**Correspondence:** O. Schneising (oliver.schneising@iup.physik.uni-bremen.de)

**Abstract.** The switch from the use of coal to natural gas or oil for the energy generation potentially reduces the greenhouse gas emissions and thus the impact on global warming and climate change because of the larger energy content per $CO_2$ molecule emitted. However, the climate benefit over coal is offset by methane ($CH_4$) leakage from natural gas and petroleum systems, which reverses the climate impact mitigation if the rate of fugitive emissions exceeds the compensation point at which the

global warming resulting from the leakage and the benefit from the reduction of coal combustion coincide. Consequently, an accurate quantification of the $CH_4$ emissions from the oil and gas industry is essential to evaluate the suitability of natural gas and petroleum as bridging fuels on the way to a carbon-neutral future.

    We show that regional $CH_4$ release from large oil and gas fields can be monitored from space by using dense daily recurrent measurements of the TROPOspheric Monitoring Instrument (TROPOMI) onboard the Sentinel-5 Precursor satellite to quantify

emissions and leakage rates. The average emissions for the time period 2018/2019 from the five most productive basins in the United States, the Permian, Appalachia, Eagle Ford, Bakken, and Anadarko are estimated to be $3.18 \pm 1.13$ Mt yr$^{-1}$, $2.36 \pm 0.88$ Mt yr$^{-1}$, $1.37 \pm 0.63$ Mt yr$^{-1}$, $0.89 \pm 0.56$ Mt yr$^{-1}$, and $2.74 \pm 0.74$ Mt yr$^{-1}$, respectively. This corresponds to $CH_4$ leakage rates relative to the associated production between $1.2\%$ and $1.4\%$ for the first four production regions, which are consistent with bottom-up estimates and likely fall below the break-even leakage rate for immediate climate benefit. For

the Anadarko Basin, the fugitive emission rate is larger and amounts to $3.9 \pm 1.1\%$, which likely exceeds the break-even rate for immediate benefit and roughly corresponds to the break-even rate for a 20-year time horizon. The determined values are smaller than previously derived satellite-based leakage rates for the time period 2009-2011, which was an early phase of hydraulic fracturing, indicating that it is possible to improve the climate footprint of the oil and gas industry by adopting new technologies and that the efforts to reduce the methane emissions have been successful. For two of the world's largest natural

gas fields Galkynysh and Dauletabad in Turkmenistan, we find collective methane emissions of $3.26 \pm 1.17$ Mt yr$^{-1}$, which corresponds to a leakage rate of $4.1 \pm 1.5\%$ suggesting that the Turkmen energy industry is not employing methane emission avoidance strategies and technologies as successfully as those currently widely used in the United States. The leakage rates in Turkmenistan and in the Anadarko Basin indicate that there is potential to reduce fugitive methane emissions from natural gas and petroleum systems worldwide. In particular, relatively newly developed oil and gas plays appear to have larger leakage

rates as compared to more mature production areas.

# 1 Introduction

Methane ($CH_4$) is an important greenhouse gas, which accounts for the second largest share of radiative forcing caused by human activities since preindustrial times. It has a much shorter atmospheric lifetime and a considerably higher global warming potential than the most important anthropogenically modified greenhouse gas carbon dioxide ($CO_2$) (Holmes et al., 2013). Hence, a combined climate change mitigation strategy, aiming at reducing both $CO_2$ and $CH_4$ emissions in parallel, addresses long-term and near-term effects of global warming and is required to achieve climate goals most efficiently (Shindell et al., 2012; Shoemaker et al., 2013).

An integral contribution to anthropogenic methane emissions originates from the exploitation of natural gas and oil for energy generation (i.e. the production of natural gas and oil, the refining of oil, and the subsequent storage, distribution and combustion of these fuels). To assess the climate impact of the production of natural gas or oil in comparison to coal, the fugitive emission rate relative to production is a key parameter. Although the combustion of natural gas or oil produces less $CO_2$ than coal at the same energy content, methane emissions during the production and distribution process offset the climate benefit over coal. Hence, there is a compensation point, the break-even rate, at which the climate impacts of gas/oil and coal coincide. The exact break-even rate depends on the time horizon, the climate impact metric (e.g. global warming potential (GWP) or technology warming potential (TWP)), and the considered fuel-switching scenario. It has been estimated that an immediate climate benefit of switching from coal-fired to gas-fired power plants requires life cycle methane emissions to stay below $3\%$ (Alvarez et al., 2012). For a time horizon of 20 years the corresponding break-even rate is about $4\%$, which drops to $2\%$ if carbon capture and sequestration becomes available (Farquharson et al., 2016).

The latest official bottom-up estimate of methane emissions from natural gas and petroleum systems reported by the United States Environmental Protection Agency (EPA) is $8.13\,\mathrm{Mt}$ $[5.12–11.54; 2\sigma]$ in 2017, corresponding to $0.9\%$ $[0.5–1.2; 2\sigma]$ of aggregated gross production (U.S. Environmental Protection Agency, 2019a), which is very likely below the break-even emission rate. As in all leakage rate estimations presented here, combined oil and gas production in terms of energy content is the reference value of the calculation and a methane content of $93\%$ in natural gas (U.S. Environmental Protection Agency, 2019b) is used to determine the mass fraction of methane in the produced natural gas. Alternative bottom-up estimates (Alvarez et al., 2018) find total U.S. oil and gas emissions of $13.0\,\mathrm{Mt}$ $[11.3–15.1; 2\sigma]$ in 2015, which is $63\%$ higher than the EPA estimate and corresponds to a fugitive emission rate of $1.4\%$ $[1.1–1.6; 2\sigma]$ (relative to combined oil and gas production).

However, several top-down studies suggest that the oil and gas industry leaks substantially more methane than assumed in official inventories, at least locally or temporally with highly variable regional leakage rates occasionally reaching several multiples of the expected bottom-up estimates (Pétron et al., 2012; Karion et al., 2013; Caulton et al., 2014; Brandt et al., 2014; Schneising et al., 2014; Peischl et al., 2015, 2016, 2018; Alvarez et al., 2018). This points to a heterogeneity of the methane leakage and complicates the specification of typical emission rates, which are necessary to reliably assess the climate footprint of the natural gas and petroleum industry as a whole.

In the past, the satellite-based detection of $CH_4$ emissions from the oil and gas industry was mostly limited to long-term averages typically yielding emission rates with associated large uncertainties (Schneising et al., 2014; Turner et al., 2016; Buchwitz

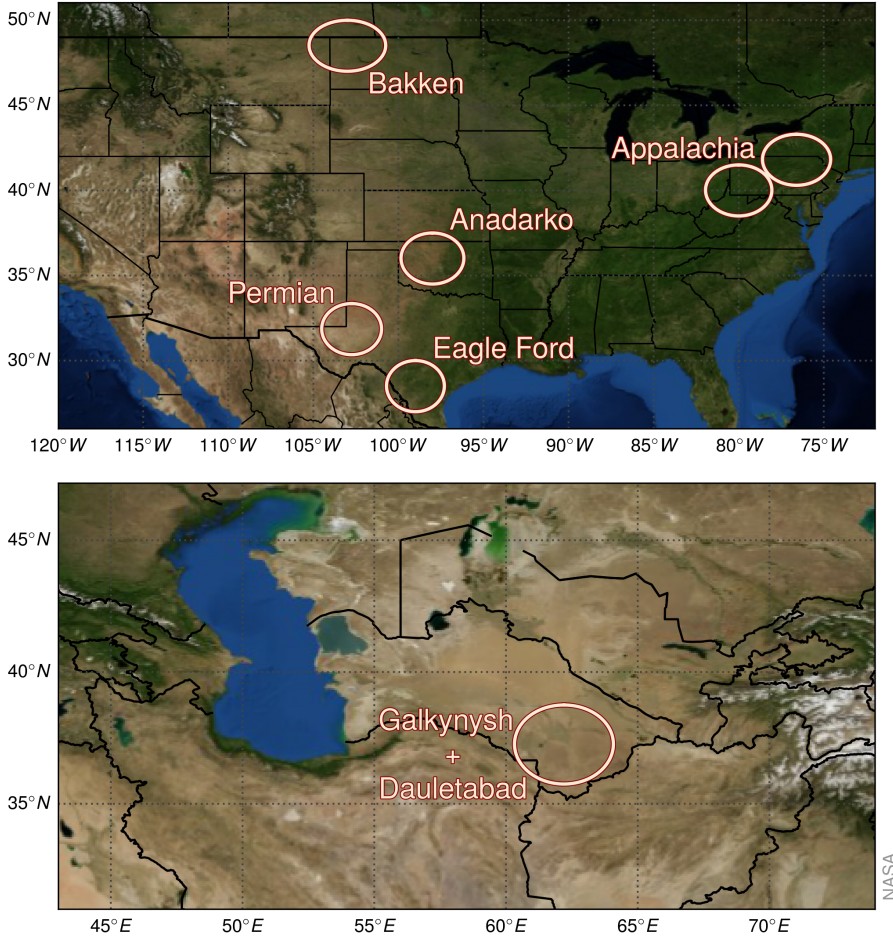

**Figure 1.** Location of the analysed natural gas and oil production regions. Due to its large and elongated extent, the Appalachia region is split into two subregions. All circles have a radius of $166.5\,\mathrm{km}$ corresponding to $1.5°$ after the coordinate transformation described in Section 2.

et al., 2017). Exceptional emissions of superemitters have also been observed in single satellite overpasses (Thompson et al., 2016). The recently launched TROPOspheric Monitoring Instrument (TROPOMI) onboard the Sentinel-5 Precursor satellite with its unique combination of high precision, accuracy, and spatiotemporal coverage (Veefkind et al., 2012) now enables the systematic detection of sufficiently large emission sources in a single satellite overpass. This has already been demonstrated for daily $CH_4$ enhancements from the energy sector for specific source regions in Northern America, Europe, and Turkmenistan (Varon et al., 2019; Schneising et al., 2019; Pandey et al., 2019; de Gouw et al., 2020). Here we use dense daily recurrent TROPOMI observations to reassess the emissions of petroleum and gas producing basins. The presented analysis includes emission estimates of the five most prolific basins in the United States as well as for two of the world's largest natural gas fields in Turkmenistan. The corresponding locations of these production regions are shown in Figure 1.

## 2 Data and Methods

In this study, atmospheric methane abundances are retrieved from the radiance measurements in the shortwave infrared (SWIR) spectral range of the TROPOMI instrument onboard the Sentinel-5 Precursor (Sentinel-5P) satellite using the latest version of the Weighting Function Modified DOAS (WFM-DOAS) algorithm (Buchwitz et al., 2006; Schneising et al., 2011) optimised to retrieve methane and carbon monoxide simultaneously (TROPOMI/WFMD v1.2) (Schneising et al., 2019).

Sentinel-5P was launched in October 2017 into a sun-synchronous orbit with an equator crossing time of 13:30. TROPOMI is a spaceborne nadir viewing imaging spectrometer measuring solar radiation reflected by the Earth in a push-broom configuration. It has a swath width of $2600\,\text{km}$ and combines high spatial resolution with daily global coverage. The nadir measurements in the SWIR have a horizontal resolution of $7 \times 7\,\text{km}^2$ and are sensitive to all altitude levels including the planetary boundary layer, which makes them well suited for the investigation of emissions from oil and gas fields. The retrieved TROPOMI/WFMD column-averaged dry air mole fractions of methane, $\text{XCH}_4$, are characterised by a random error (precision) of $14.0\,\text{ppb}$ and a systematic error (relative accuracy) of $4.3\,\text{ppb}$ after quality filtering (Schneising et al., 2019).

The methane emission estimation is based on daily TROPOMI observations and a Gaussian integral method. For a fixed source region, quality-filtered daily $\text{XCH}_4$ retrievals are automatically processed as described below. First the data given on geographical longitude and latitude is transformed to rotated coordinates, so that zero meridian and equator pass through the centre of the analysed region with zonal direction matching the mean wind direction. The mean wind is defined as the average of all boundary layer winds within the region (as defined by the circles in Figures 1, 4, 5, 7, 8, 10, and 11) and within the time window between 11 and 13 h local time to take the wind history into account. Thereby, the boundary layer wind at a given time and place is defined as the pressure-weighted mean of winds for all layers within the boundary layer as obtained from the hourly European Centre for Medium-Range Weather Forecasts (ECMWF) ERA5 reanalysis product (Hersbach et al., 2018). The corresponding transformation of the coordinates and wind components is described in detail in Doms and Baldauf (2018). Similar approaches based on rotating individual satellite observations according to wind direction have also been used in the analysis of other atmospheric species (Valin et al., 2013; Pommier et al., 2013; Fioletov et al., 2015).

After rotating the coordinate system, the transformed daily data is gridded on a $0.05° \times 0.05°$ grid and boxes with $\text{XCH}_4$ below the 10th percentile within a radius of $700\,\text{km}$ around the pivotal point are additionally excluded due to potential residual cloud cover that may occur occasionally (Schneising et al., 2019). The rotated coordinates have the following advantages: 1) the new grid is nearly rectangular leading to an (almost) homogeneous distribution of grid points, 2) it is straightforward to compute the integral perpendicular to wind direction, which is needed in our flux estimation, and 3) multiple daily gridfiles can be easily combined to long-term averages, because the wind always has the same orientation (left to right).

After subtracting a mean background upwind of the source, the data look like the example shown in Figure 2. The corresponding background region, which is highlighted in the figure, has the same position for all days and all investigated regions in the transformed coordinate system and its suitability to enable a reliable emission estimate of the source region for a given day is automatically evaluated using certain selection criteria introduced below (see also Section 3.7 for an assessment of the exclusionary power of the filter criteria including those concerning the background region). Let $E$ be the total column en-

**Figure 2.** Example daily data to illustrate the estimation of the emission. The coordinate system has been transformed so that the wind direction lies along the equator. The background region is shown in skyblue, the plume region in orange, and the hot spot region in pink. The extent of these regions is fixed for all analysed days in the rotated coordinate system. By design, the background region has a mean abundance of $0\,\mathrm{ppb}$. The meridional sections used to calculate the daily flux are displayed in red.

hancement (in units of mass per area) and $v$ the mean boundary layer wind speed. To estimate the daily emission rate $\Phi$, we calculate fluxes of the vector field $E\boldsymbol{v}$ through cross-sections perpendicular to wind direction (meridional red lines in Figure 2) according to the divergence theorem; the flux through the other three sides of a rectangle surrounding the source region is assumed to be negligible (unit normals of the zonal boundaries are perpendicular to wind direction and the upwind meridional boundary is in the background with $E = 0$). The flux through the $k$th cross-section is

$$\Phi_k = \int\limits_V (\boldsymbol{\nabla} \cdot E\boldsymbol{v})\,dV = \oint\limits_{\partial V = S} E\boldsymbol{v} \cdot d\boldsymbol{S} = \sum_i E_i\, v\,\Delta l_i = v\,\Delta l \sum_i E_i = \frac{v \cdot \Delta l \cdot M_{\mathrm{CH_4}} \cdot \rho_{dry}}{N_A \cdot A_{\mathrm{CH_4}}} \sum_i (\Delta \mathrm{XCH_4})_i \quad . \tag{1}$$

Thereby, $\Delta l$ is the size of a grid box ($0.05°$ is equivalent to about $5\,\mathrm{km}$ near the equator) and $i$ corresponds to meridional summation along the $k$th red line. The molar mass of methane $M_{\mathrm{CH_4}} = 16.04\,\mathrm{g\,mol^{-1}}$, the Avogadro constant $N_A = 6.022 \cdot 10^{23}\,\mathrm{molec\,mol^{-1}}$, and the mean dry air column $\rho_{dry}$ (in units of molecules per area) within a radius corresponding to $3°$ after the coordinate transformation are used to convert between the enhancement in $\mathrm{XCH_4}$ and the total column enhance-

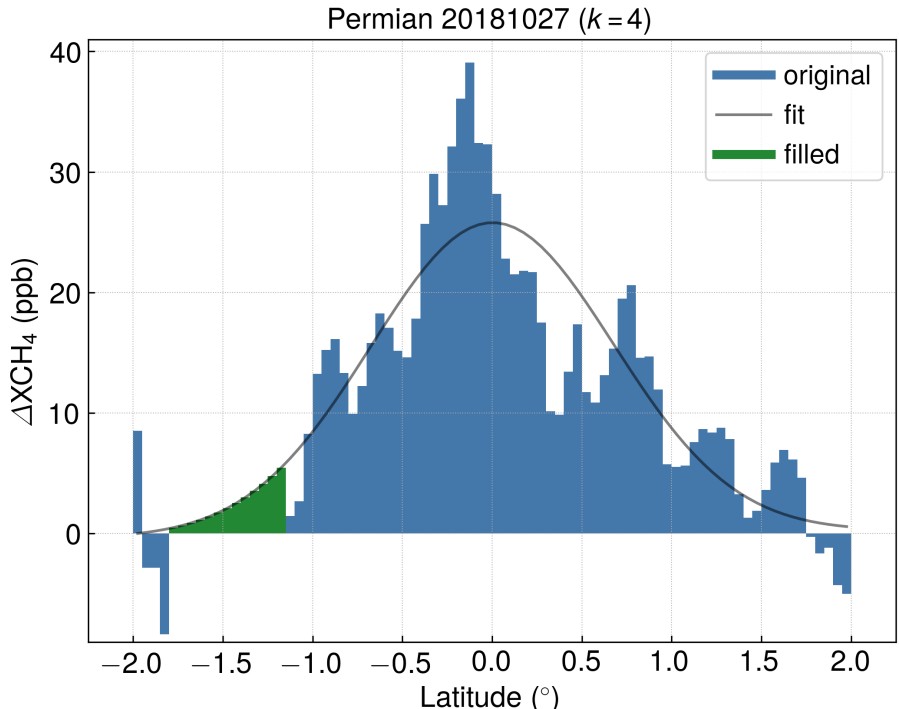

**Figure 3.** Demonstration of the gap-filling procedure. For $\Phi_k$ with at least $60\%$ of all maximum available grid boxes along the meridional section, the gaps in the original data (blue) are filled (green) according to a fitted linear combination of a Gaussian and a linear polynomial (grey). The $\Phi_k$ with less than $60\%$ of data are not used in the estimation of the daily emissions. Shown here are the results for a specific $k$ on an example day.

ment $E$; $A_{CH_4}$ is the dimensionless near-surface averaging kernel (which is about $1.02$ for all source regions analysed here) characterising the boundary layer sensitivity of the retrieval valid for the present mean altitude.

The average over all $\Phi_k$ then yields the final daily flux estimate $\Phi$. Thereby, all $\Phi_k$ with at least $60\%$ of all maximum available grid boxes existing along the $k$th meridional section are selected. If there are no such $\Phi_k$ to average, there is no flux estimate for this specific day. Before averaging, the gaps of the selected $\Phi_k$ are filled according to a fitted linear combination of a Gaussian and a linear polynomial (see Figure 3 for an example meridional section).

The corresponding total $1\sigma$-uncertainty $u_\Phi$ is determined by the individual uncertainty components relative to the respective means via

$$\left(\frac{u_\Phi}{\Phi}\right)^2 = \frac{u_{v,\mathrm{abs}}^2 + u_{v,\mathrm{dir}}^2}{v^2} + \left(\frac{u_{\rho_{dry}}}{\rho_{dry}}\right)^2 + \left(\frac{u_E}{E}\right)^2 \qquad (2)$$

with $u_{v,\mathrm{abs}}$ being the standard deviation of all absolute boundary layer wind speed values over the selected region between 11 and 13 h local time and $u_{v,\mathrm{dir}}$ quantifying the uncertainty due to the maximal mean wind direction change in the considered 2 hour time window of wind history; $u_{\rho_{dry}}$ is the standard deviation of the dry air columns within the same region used

to determine the mean value, and $u_E$ is the standard deviation of the enhancement integrals along the different meridional sections. For the regions under consideration in this study, the impact of topography (via $u_{\rho_{dry}}$) is small compared to the other parameters contributing to the flux uncertainty (see Section 3.7).

Of all available days, those with sufficient amount of data are selected to calculate the averaged long-term emission rate $\bar{\Phi}$ of the corresponding source region and the regional mean enhancement distribution. Thereby, at least $25\%$ of the background, plume, and hot spot regions (see Figure 2) must be filled with data after quality filtering in each case and there have to be at least four cross-sections with more than $60\%$ of data to average. As scenes with low wind speed may be dominated by diffusion and scenes with large wind velocity exhibit smaller column enhancements, additional criteria require that $v \in (2\,\mathrm{m\,s^{-1}}, 10\,\mathrm{m\,s^{-1}})$. Moreover, the enhancement distribution is required to be sufficiently uniform with respect to the equator and in the background ($|\bar{E}_b^N - \bar{E}_b^S| < 10\,\mathrm{ppb}$, $|\bar{E}_p^N - \bar{E}_p^S| < 15\,\mathrm{ppb}$, $\sigma(E_b) < 10\,\mathrm{ppb}$) to minimise scenarios with potential residual cloudiness or considerably wrong wind direction, where $\bar{E}_{b,p}^{N,S}$ is the mean enhancement on the northern/southern half of the background/plume region and $\sigma(E_b)$ is the standard deviation of the enhancements in the background region. Furthermore, days with mean wind direction changes during the considered 2 hour time window that are larger than $30°$ or with uncertainty estimates larger than $5\,\mathrm{Mt\,yr^{-1}}$ are additionally excluded. The corresponding uncertainty $u_{\bar{\Phi}}$ of the mean long-term emission rate is given via error propagation by the root sum square of the individual daily uncertainties $u_\Phi$ divided by the number of effectively contributing days $n_{\mathrm{eff}}$, which is smaller than the actual number of days due to expected correlation of neighbouring data points,

$$u_{\bar{\Phi}} = \frac{\sqrt{\sum_j u_{\Phi,j}^2}}{n_{\mathrm{eff}}} \quad . \tag{3}$$

We assume uncorrelated data blocks with a length of one month, i.e. $n_{\mathrm{eff}}$ is the number of months containing emission estimates contributing to the mean.

The associated long-term leakage rate is then calculated by normalising the estimated emissions $\bar{\Phi}$ by combined oil and gas gross production of the considered region in terms of energy content (Schneising et al., 2014). In order to express the leakage rate as a percentage, emission (dividend) and production (divisor) are converted to the same units (energy per time, see Table 1). To quantify the combined production, the natural gas is converted to barrel of oil equivalents (BOE) by using a factor depending on its energy content. Although the exact conversion factor varies slightly with the specific composition of the natural gas, we use the widely used relationship of 6000 cubic feet per BOE (U.S. Energy Information Administration, 2019). To convert between emitted $CH_4$ mass and natural gas volume in cubic feet via the ideal gas law, we assume standard natural gas reference conditions (International Organization for Standardization, 1996) ($T = 288.15\,\mathrm{K}$, $p = 1013.25\,\mathrm{hPa}$) and a $CH_4$ content of $93\%$ in natural gas (U.S. Environmental Protection Agency, 2019b) with a realistic range of $87–99\%$ for high caloric gas. Please note that all comparative leakage rates presented here (e.g. the bottom-up and airborne-based estimates) were also calculated as or converted to combined energy loss rates to make them comparable to our estimates.

For a production mixture of oil and gas the break-even rate of about $3\%$ estimated in the literature for gas-only production (Alvarez et al., 2012; Farquharson et al., 2016) has to be reduced as oil produces more $CO_2$ per unit of energy than natural gas. The fuel-related emission factors for bituminous coal, crude oil, and natural gas are 95, 73, and $56\,\mathrm{tCO_2\,TJ^{-1}}$ (Intergovernmental Panel on Climate Change, 2006). Thus, oil has about $56\%$ of the emission saving potential of natural gas when replacing

coal. As the maximal share of oil in the production mixtures of the analysed production regions is $75\%$ (see Section 3 and Table 1), the smallest occurring compensation value of all considered mixtures is assumed to be $(0.75 \cdot 0.56 + 0.25) \cdot 3\% = 2\%$. Consequently, we assume a break-even range of $2$–$3\%$ to achieve immediate climate benefit when switching from coal-based to a typical mixture of gas- and petroleum-based energy generation for a natural gas share between $25\%$ and $100\%$.

## 3 Results and Discussion

The top five producing basins in the United States during 2018/2019 in order of combined oil and gas production in terms of energy content were the Permian, Appalachia, Eagle Ford, Bakken, and Anadarko (U.S. Energy Information Administration, 2020), which are therefore potential candidates for our approach. As rotation in wind direction is an important element in the presented method, nearly rotation-symmetrical basins such as the Permian, Bakken, and Anadarko are best suited for the analysis. Due to its large and elongated extent, the Appalachia region is split into two subregions. The tubular shape of Eagle Ford with almost linearly arranged sources and its proximity to the offshore sources in the Gulf of Mexico complicate the analysis for this region. We also consider Galkynysh and Dauletabad, which are two of the world's largest natural gas fields located in Turkmenistan, to put the American results into a global context.

### 3.1 Permian

The Permian is a sedimentary basin in western Texas and eastern New Mexico. It has become one of the most productive oil producing regions in the world and is by far the most prolific oil field in the United States. In the recent past, the share of natural gas is increasing as wells get older and fewer new wells are drilled. The average production in the period 2018/2019 was 3897 thousand barrels of oil (Mbbl) and 13182 million cubic feet (MMcf) natural gas per day (U.S. Energy Information Administration, 2020) corresponding to a total combined energy production of $6094\,\text{kBOE}\,\text{d}^{-1}$ (kilo barrel of oil equivalent per day). Thus, the production mix consists of about $65\%$ oil and $35\%$ natural gas. The Permian is subdivided in two major lobes with unconventional oil and gas production, the Delaware and the Midland Basin, which are separated by the Central Basin Platform dominated by conventional production. The qualitative detection of daily methane enhancements for the Permian has recently been demonstrated using TROPOMI measurements (de Gouw et al., 2020). Usually, the methane emissions of the Delaware and Midland Basins are detected independently in the daily satellite data (for example in Figure 2). The averaged enhancement distribution for the period 2018/2019 and the daily emission estimates are shown in Figure 4. The associated mean emission estimate for this period is $3.18 \pm 1.13\,\text{Mt}\,\text{yr}^{-1}$ corresponding to a fugitive emission rate of $1.3 \pm 0.5\%$ relative to combined oil and gas energy production, which is slightly larger than the national bottom-up estimate of the EPA ($0.9\%$ $[0.5$–$1.2; 2\sigma]$) but consistent with Alvarez et al. (2018) ($1.4\%$ $[1.1$–$1.6; 2\sigma]$) and likely below the break-even leakage rate for immediate climate benefit.

Concurrent with our study, Zhang et al. (2020) also quantified methane emissions from the Permian basin using a different data set and an alternative inversion method combining information from the operational TROPOMI methane product and prior emission estimates within a Bayesian framework. Despite these quite distinct approaches, their total emission estimate of

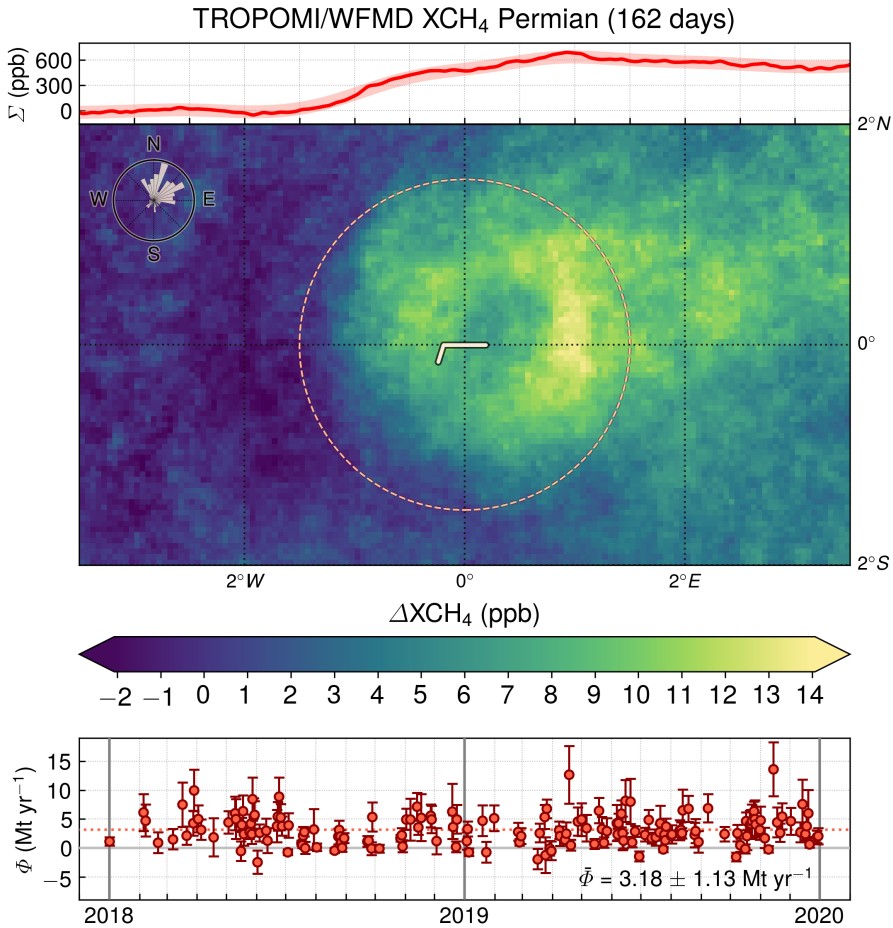

**Figure 4.** Averaged enhancement distribution with associated integrated enhancement along the meridional sections (top) and daily emission estimates $\Phi$ (bottom) for the Permian Basin. The coordinates of the pivotal point are $31.85°$N and $102.75°$W; the radius of the circle highlighting the approximate basin extent is $166.5$ km corresponding to $1.5°$ after the coordinate transformation. The daily $\Phi$ are used to determine the mean emission $\bar{\Phi}$, which corresponds to a fugitive emission rate of $1.3 \pm 0.5\%$. The distribution of the associated original daily wind directions (defined as the direction where the wind blows to) can be seen in the overlayed wind rose in the upper left corner.

$2.9 \pm 0.5\,\mathrm{Mt\,yr^{-1}}$ based on satellite observations from May 2018 to March 2019 agrees within uncertainties with our estimate. If we restrict our analysis to this specific period, the consistency becomes even better and we get the almost identical estimate of $2.8\,\mathrm{Mt\,yr^{-1}}$ with our method, which is independent of prior knowledge. Therefore, the corresponding absolute results are considered very robust. However, there is a crucial difference in the calculation and subsequent interpretation of the leakage
rate: while our rate ($1.3\%$) is calculated relative to combined oil and gas production in terms of energy content (Schneising et al., 2014), the rate of Zhang et al. (2020) is larger ($3.7\%$) and appears more alarming because it is put in relation to natural gas production only. With this alternative divisor we would also get a leakage rate of $3.7\%$ (as can be determined from Table 1). But as the Permian is dominated by oil production, we consider the total energy approach to be better suited to assess the climate impact compared to coal in general. Otherwise, the energy content of the extracted oil would be neglected and a pure
oil play (with an infinitesimal fraction of not marketed but vented natural gas) would have a leakage rate of $100\%$. For a pure natural gas play, however, both approaches to determine the leakage rate coincide.

## 3.2   Bakken

The Bakken formation is the second largest oil-producing region in the United States with production mainly concentrated on North Dakota. The oil and natural gas in the Bakken are locked in rock reservoirs with low permeability and unconventional
drilling methods were necessary to transform the formation into a prolific production region. The production mix consists of about $75\%$ oil and $25\%$ natural gas and the average production in the period 2018/2019 was $1361\,\mathrm{Mbbl}$ oil and $2661\,\mathrm{MMcf}$ natural gas per day (U.S. Energy Information Administration, 2020) corresponding to a total combined energy production of $1805\,\mathrm{kBOE\,d^{-1}}$. The determined mean emission estimate for the period 2018/2019 is $0.89 \pm 0.56\,\mathrm{Mt\,yr^{-1}}$. As can be seen in Figure 5, the associated averaged enhancement distribution is noisier due to less contributing days and does not show a plume
structure as clear as in the case of the Permian. Together with the large relative uncertainty, this suggests that the emissions of the Bakken are close to the detection limit for daily data. The estimated emissions correspond to a fugitive emission rate of $1.3 \pm 0.8\%$ relative to combined oil and gas energy production, which is consistent with Alvarez et al. (2018) and slightly larger than the national EPA bottom-up estimate. The rate is likely below the break-even rate but the error bars extend in the break-even range. The derived leakage rate is also consistent with the energy loss rates of $1.6 \pm 0.5\%$ and $1.4 \pm 0.5\%$
estimated for Bakken from airborne data taken in May 2014 (Peischl et al., 2016) and April 2015 (Peischl et al., 2018). The respective airborne-based estimates were originally specified as leakage rates relative to natural gas production only and have been converted to rates relative to combined oil and gas production in terms of energy content in each case by considering the natural gas fraction of $25\%$ in the production mix of the Bakken to make the estimates directly comparable to our estimates. The Bakken estimate from this study is smaller than previously derived satellite-based leakage rates for the time period 2009-2011,
which were estimated to be $10.1 \pm 7.3\%$ for this early phase of hydraulic fracturing (Schneising et al., 2014). Although the corresponding uncertainties of both satellite studies for Bakken are large, the reduction of relative leakage over time suggests that the climate footprint of the oil and gas industry can be improved by adopting new technologies and that the efforts to reduce fugitive methane emissions have been successful. The systematic measures implemented proactively by coalitions of oil and gas companies since 2014 to continuously reduce methane emissions include additional leak detection and repair campaigns,

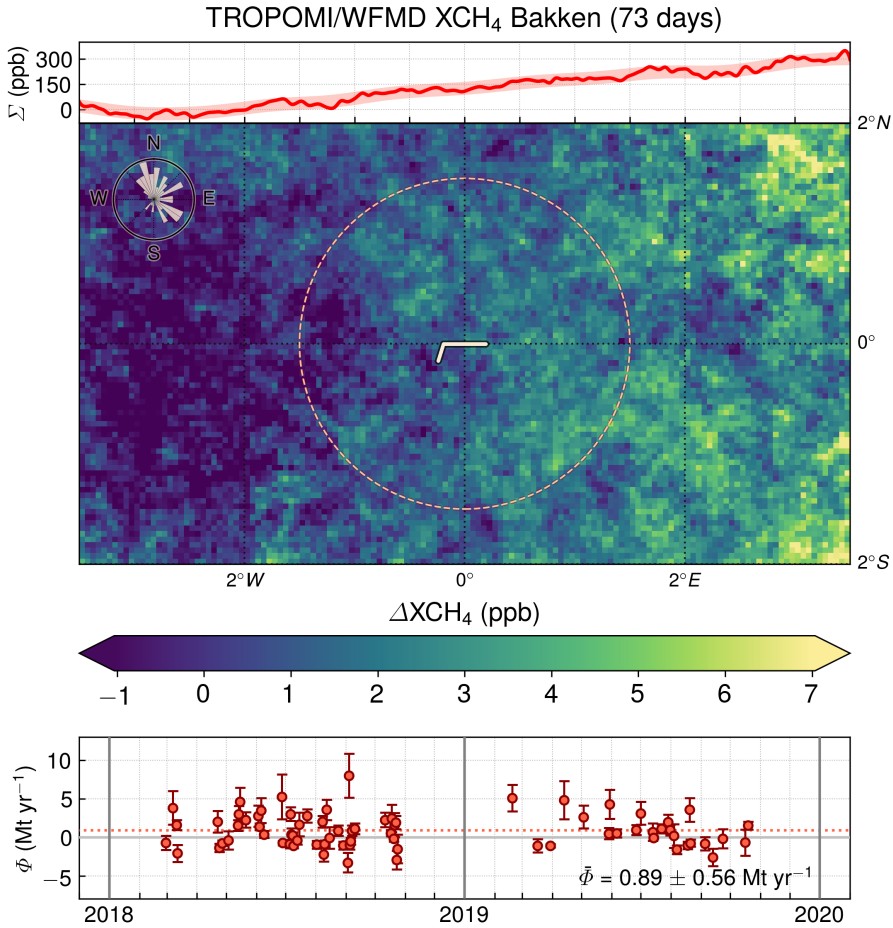

**Figure 5.** As Figure 4 but for the Bakken formation. The coordinates of the pivotal point are $48.5°$N, $103°$W. The mean emission estimate $\bar{\Phi}$ corresponds to a fugitive emission rate of $1.3 \pm 0.8\%$.

replacement or upgrade of high-emitting devices, and reduction of venting or flaring, to direct toward the ambitious goal of achieving a leakage rate not exceeding $1\%$ across the natural gas supply chain (including a maximum of $0.3\%$ from upstream operations) by 2025 (ONE Future, 2019; Oil and Gas Climate Initiative, 2019). An illustration of the decreasing leakage rates derived from satellite and airborne measurements in the discussed publications is shown together with the assumed break-even

5    range for immediate climate benefit in Figure 6.

### 3.3 Appalachia

The Appalachia region in eastern North America consists of several stacked formations, most prominently the Marcellus and Utica shale plays. The region drives the overall increase in the United States natural gas production. The Marcellus shale is a unit of sedimentary rock and the most productive natural gas-producing formation in the Appalachian Basin. The older Utica

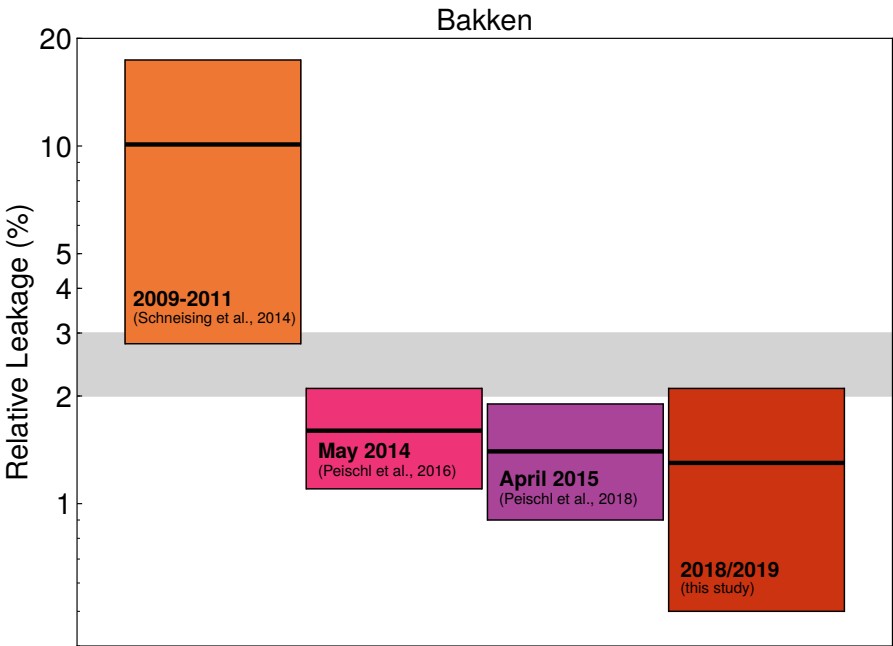

**Figure 6.** Comparison of fugitive emission rates for the Bakken formation from different studies with a reduction of relative leakage over time. See main text for details. The assumed break-even range for immediate climate benefit is shown in grey.

and Point Pleasant formations extend below the Marcellus shale and the Upper Devonian shale above it. Unconventional drilling in the Appalachian is mainly focused on two hot spot regions in the southwestern and northeastern part of Pennsylvania. These two regions are analysed separately due to the large and elongated extent of the basin. The average production of the Appalachia region during 2018/2019 was 30312 MMcf natural gas and 127 Mbbl oil per day (U.S. Energy Information Administration,
5  2020) corresponding to a total combined energy production of $5179\,\mathrm{kBOE\,d^{-1}}$. Thus, the production mix is strongly gas driven with 98% natural gas and only 2% oil. The averaged enhancement distribution during 2018/2019 for the southwestern part of the Appalachia region and the daily emission estimates are shown in Figure 7. The associated mean emission estimate is $1.07 \pm 0.45\,\mathrm{Mt\,yr^{-1}}$. The corresponding estimation for the northeastern part provides $1.29 \pm 0.43\,\mathrm{Mt\,yr^{-1}}$. Thus, the mean emission for the complete Appalachian region amounts to $2.36 \pm 0.88\,\mathrm{Mt\,yr^{-1}}$. However, it has to be noted that there are only
10  a few days contributing to this emission estimate, namely 24 days for the southwestern and 10 days for the northeastern part of the Appalachian. The derived emissions are equivalent to a fugitive emission rate of $1.2 \pm 0.4\,\%$, which is consistent with the bottom-up estimates and likely below the compensation point for a climate benefit on all time frames. The leakage rate inferred from the satellite measurements of $\mathrm{XCH_4}$ is higher than the loss rate of $0.3 \pm 0.1\,\%$ estimated for the northeastern Marcellus from airborne data taken in July 2013 (Peischl et al., 2015). Although this airborne estimate was specified as a leakage rate
15  relative to natural gas production only, it can be compared directly to our estimate because the Appalachian region is strongly

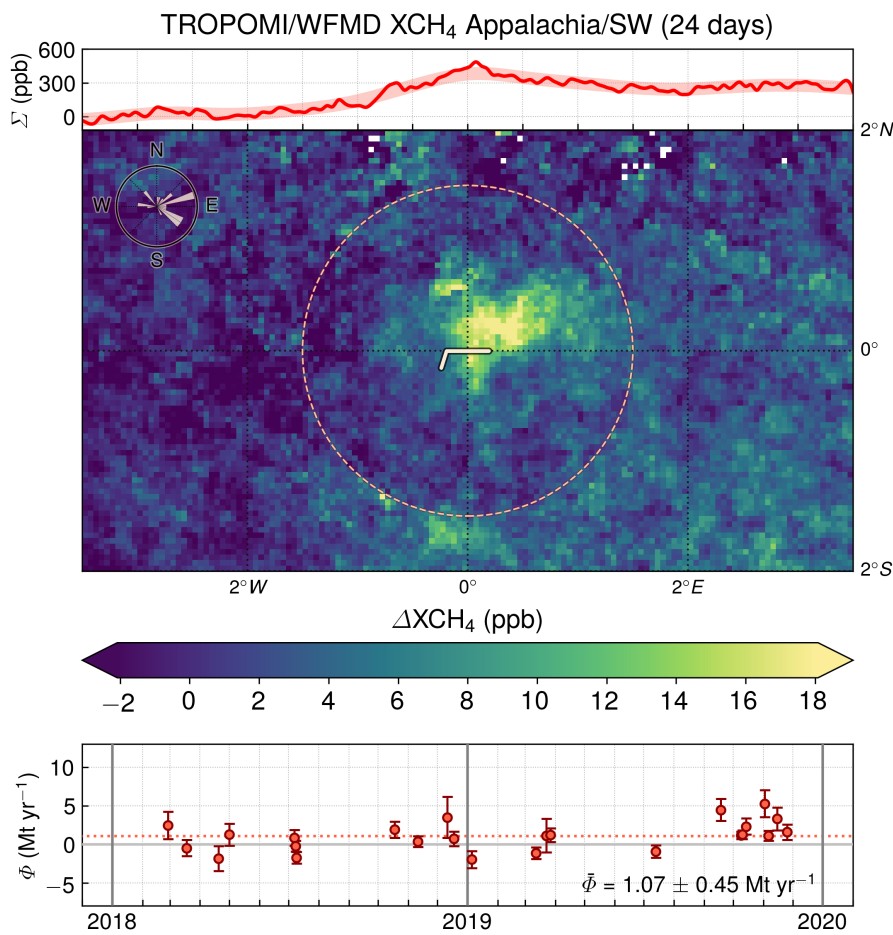

**Figure 7.** As Figures 4 and 5 but for the southwestern part of the Appalachia region. The coordinates of the pivotal point are $40°$N, $80°$W. The pivotal point of the second subregion not shown here is located at $41.8°$N, $76.6°$W.

gas driven ($98\%$) and the conversion to a rate relative to combined oil and gas production in terms of energy content has thus only a marginal impact.

### 3.4 Eagle Ford

The Eagle Ford Shale is a geological formation in southern Texas, which extends from the Mexican border to the northeast
5  in a tubular shape. The brittleness of the rock in the high-carbonate areas in the western part of the formation makes it more conducive to hydraulic fracturing and a lot of capital has been invested to develop its unconventional hydrocarbon extraction. The production mix consists of about $55\%$ oil and $45\%$ natural gas and the average production in the period 2018/2019 was $1344\,\text{Mbbl}$ oil and $6674\,\text{MMcf}$ natural gas per day (U.S. Energy Information Administration, 2020) corresponding to a total combined energy production of $2456\,\text{kBOE}\,\text{d}^{-1}$. Due to its shape with almost linearly arranged sources and the proximity to

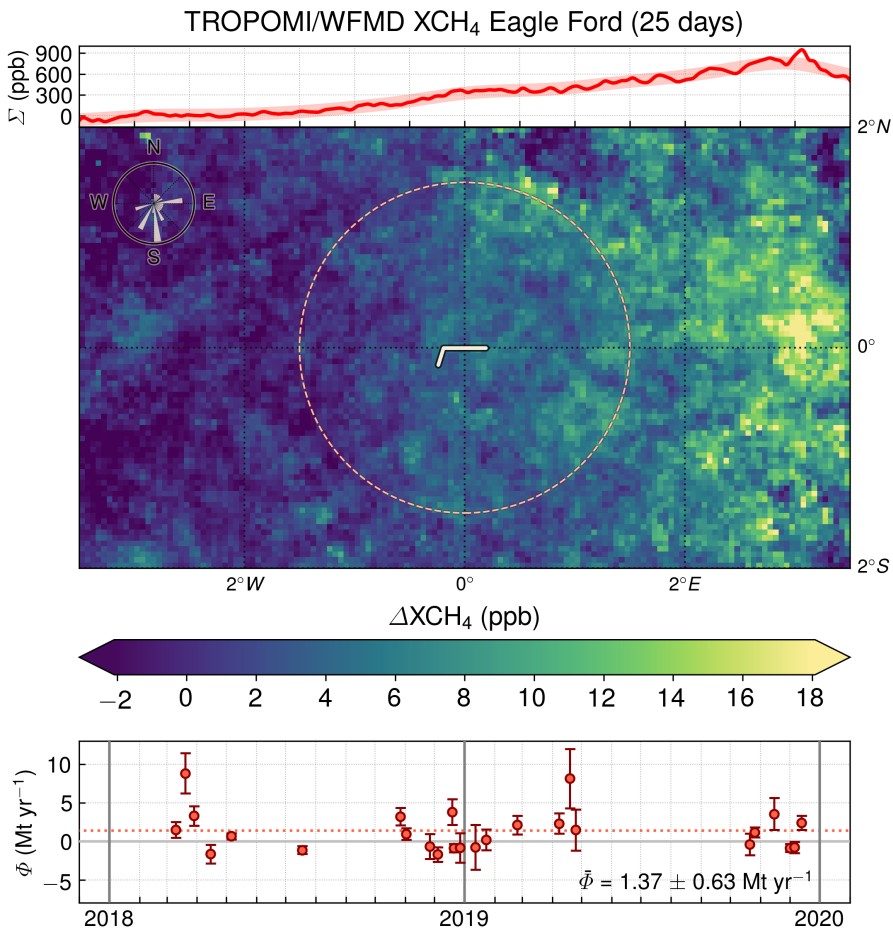

**Figure 8.** As Figures 4, 5, and 7 but for the Eagle Ford formation. The coordinates of the pivotal point are $28.5°$N, $99°$W. To avoid that the background is impacted by offshore sources, wind directions blowing towards the northwest were additionally excluded.

significant offshore sources in the Gulf of Mexico (Yacovitch et al., 2020), the introduced approach is challenging in the case of Eagle Ford. To avoid that the background is impacted by the offshore sources, wind directions blowing towards the northwest (between North and West) were additionally excluded. As a consequence, almost all summer days, where the winds mainly blow off the sea, are filtered out, leaving only 25 days for analysis (see Figure 8). The associated mean emission estimate for

5    the period 2018/2019 is $1.37 \pm 0.63 \, \mathrm{Mt\,yr^{-1}}$ corresponding to a leakage rate of $1.4 \pm 0.7\,\%$, which is consistent with Alvarez et al. (2018) and slightly larger than the national EPA bottom-up estimate. Although the estimated fugitive emission rate is below $2\,\%$, the error bars extend in the break-even range. The derived leakage rate is also consistent with estimates based on airborne data taken in April 2015 (Peischl et al., 2018), which report a mean loss rate of $2.6 \pm 0.9\,\%$ relative to natural gas production. This corresponds to an energy loss rate of $1.2 \pm 0.4\,\%$ when taking the natural gas share of $45\,\%$ in the production

10    mix into account. As in the case of the Bakken, the Eagle Ford estimate from this study is smaller than previously derived

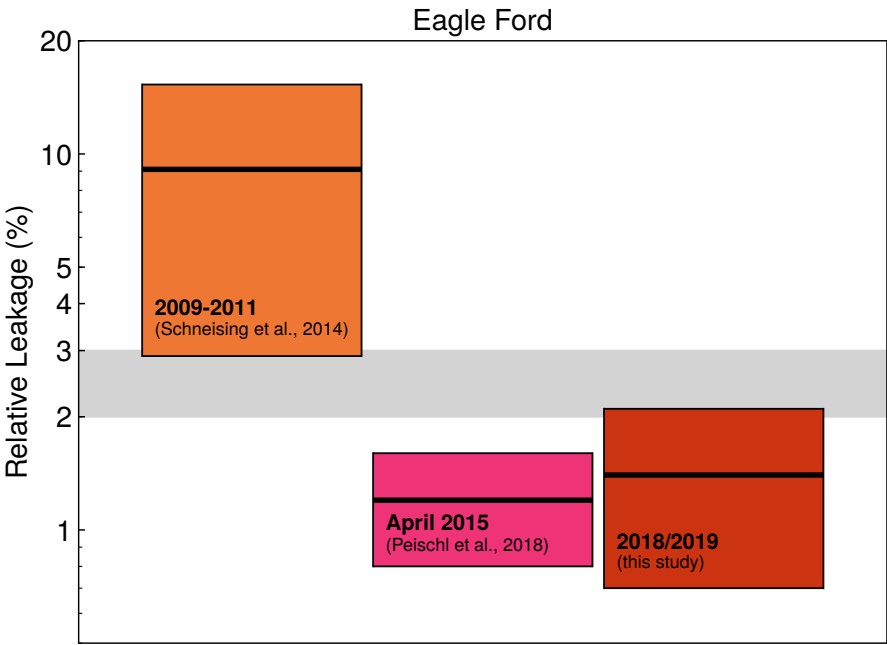

**Figure 9.** Comparison of fugitive emission rates for the Eagle Ford Shale from different studies with a reduction of relative leakage over time. See main text for details. The assumed break-even range for immediate climate benefit is shown in grey.

satellite-based leakage rates for the time period 2009-2011, which were estimated to be $9.1 \pm 6.2\,\%$ (Schneising et al., 2014), suggesting that the emissions have been reduced by improving the technological standards since the early phase of hydraulic fracturing. An illustration of the decreasing leakage rates derived from satellite and airborne measurements in the discussed publications is shown together with the assumed break-even range for immediate climate benefit in Figure 9.

## 3.5 Anadarko

The Anadarko Basin is located in the western part of Oklahoma and the bordering states. It is one of the most prolific natural gas production regions in North America and is just beginning to exploit its unconventional production potential. It is an attractive target for the operators as it contains many stacked plays overlapping in large parts of the basin allowing for accessing multiple targets from one well pad. The average production in the period 2018/2019 was $7421\,\mathrm{MMcf}$ natural gas and $548\,\mathrm{Mbbl}$ oil per day (U.S. Energy Information Administration, 2020), which corresponds to a total production of $1785\,\mathrm{kBOE\,d^{-1}}$ and a production mix of about $70\,\%$ natural gas and $30\,\%$ oil. The averaged enhancement distribution for the period 2018/2019 and the daily emission estimates are shown in Figure 10. The mean emission estimate of $2.74 \pm 0.74\,\mathrm{Mt\,yr^{-1}}$ corresponds to a leakage rate of $3.9 \pm 1.1\,\%$, which is considerably larger than for the other analysed production regions in the United States and likely exceeds the break-even rate for immediate climate benefit. It rather corresponds to the break-even rate for a 20-year time horizon and a scenario without carbon capture and sequestration (Farquharson et al., 2016).

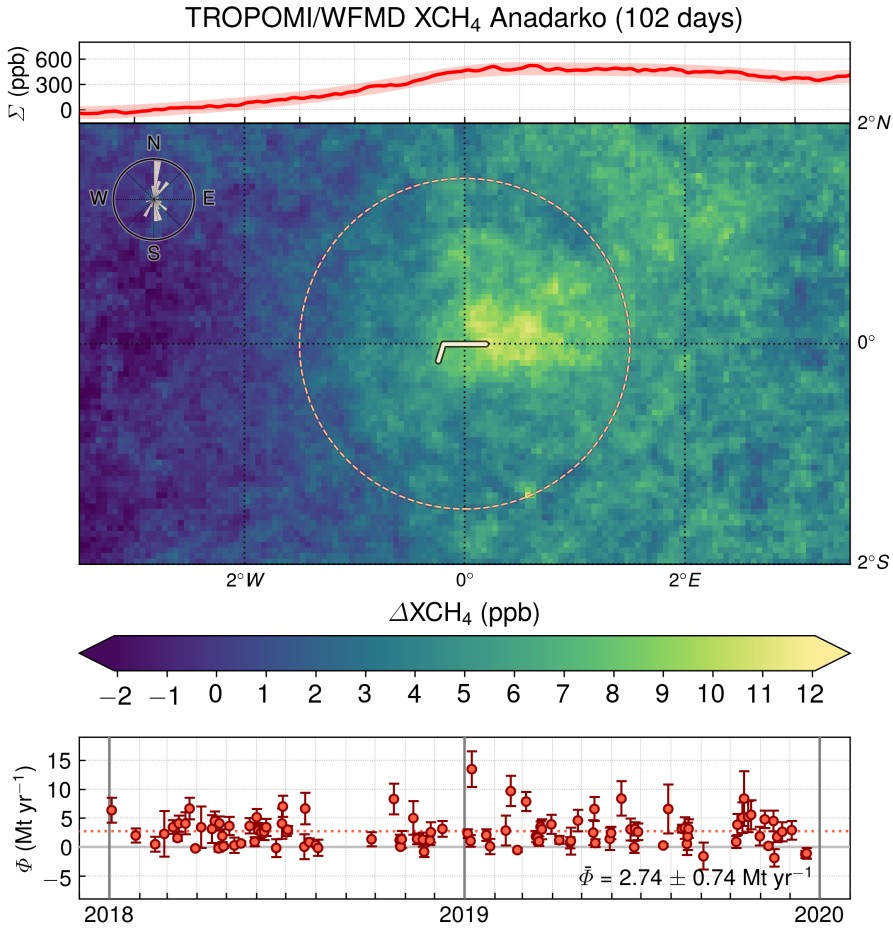

**Figure 10.** As Figures 4, 5, 7, and 8 but for the Anadarko Basin. The coordinates of the pivotal point are $36°N$, $98°W$. The mean emission estimate $\bar{\Phi}$ corresponds to a fugitive emission rate of $3.9 \pm 1.1\,\%$.

## 3.6 Galkynysh and Dauletabad

Besides the discussed production regions in the United States, we also analysed methane emissions from two of the world's largest natural gas fields, Galkynysh and Dauletabad in Turkmenistan. Galkynysh is a cluster of conventional oil and gas deposits that have been combined under a common name. Galkynysh started production in 2013 and was planned to be developed in three stages by adding capacities of about 2900, 2900, and 3400 MMcf per day in the respective stages (AidData, 2013; U.S. Energy Information Administration, 2016). As the last stage was expected to start in late 2015, we assume that the field production was close to the envisaged total quantity of 9200 MMcf per day in the analysed period 2018/2019, although official production figures are not available. Dauletabad is also a large natural gas field in Turkmenistan, which is located between Galkynysh and the Iranian border, with assumed gas production of 2900 MMcf per day (Mammadov, 2015). This corresponds

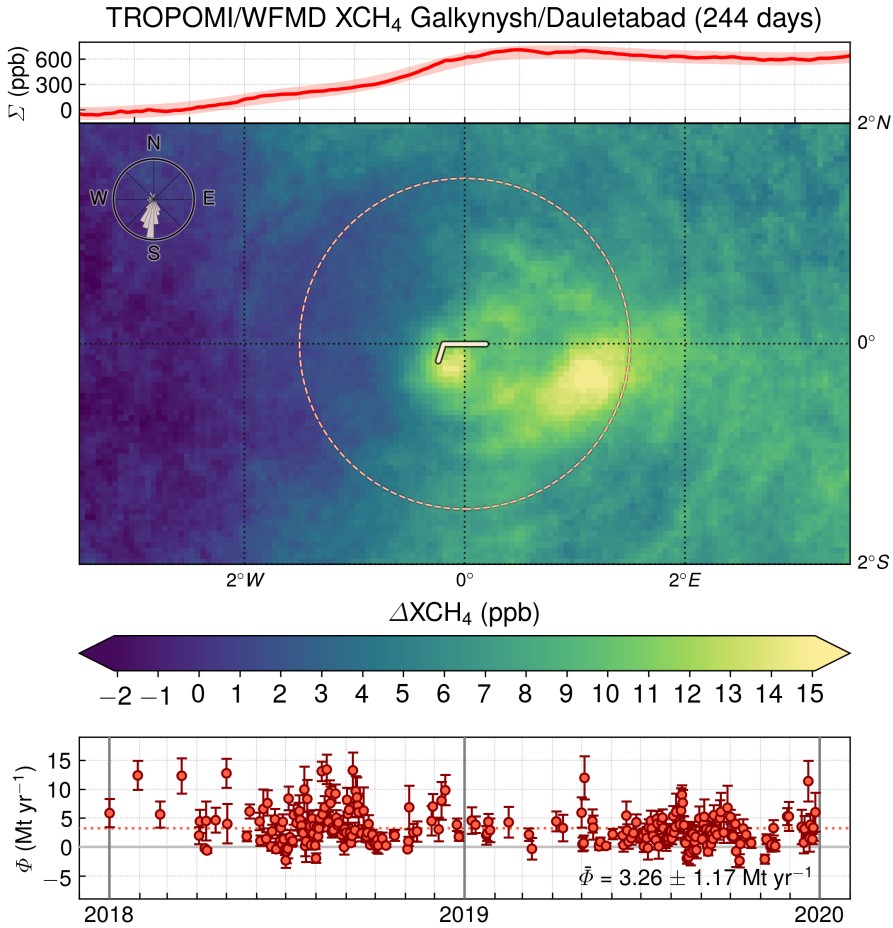

**Figure 11.** As Figures 4, 5, 7, 8, and 10 but for the Galkynysh and Dauletabad fields in Turkmenistan. The coordinates of the pivotal point are $37.25°$N, $62.2°$E.

to a total collective production of Galkynysh and Dauletabad of $2017\,\mathrm{kBOE}\,\mathrm{d}^{-1}$ because there is no reference to commercial oil production from both fields. The qualitative detection of daily methane enhancements for the Galkynysh field has already been demonstrated using TROPOMI measurements (Schneising et al., 2019). The quantitative reinforcement in this study (see Figure 11) provides a joint emission estimate together with Dauletabad of $3.26 \pm 1.17\ \mathrm{Mt\,yr}^{-1}$ corresponding to a fugitive emission rate of $4.1 \pm 1.5\,\%$, which is comparable to the leakage rate for the Anadarko basin and roughly corresponds to the break-even rate for a 20-year time horizon (Farquharson et al., 2016). Due to the lack of official production reporting, it is possible that the actual leakage rate for these Turkmen fields may be somewhat smaller (e.g. when there is also some oil production) or larger (e.g. when the targeted production of Galkynysh was not achieved in 2018/2019).

**Table 1.** Summary of the emission and production values used to determine the leakage rates (emissions divided by combined oil and gas production). All values have been converted to $\mathrm{kBOE\,d^{-1}}$ as described in Section 2. Also shown are the mean percentage variance contributions to the emission estimates for the relative uncertainty components of Equation 2.

| Region | Emissions $(\mathrm{kBOE\,d^{-1}})$ | Production | | | | | Leakage $(\%)$ | Variance contributions $(\%)$ | | | |
|---|---|---|---|---|---|---|---|---|---|---|---|
| | | Oil $(\mathrm{kBOE\,d^{-1}})$ | Gas $(\mathrm{kBOE\,d^{-1}})$ | Oil $(\%)$ | Gas $(\%)$ | Oil+Gas $(\mathrm{kBOE\,d^{-1}})$ | | $E$ | $v,\mathrm{abs}$ | $v,\mathrm{dir}$ | $\rho_{dry}$ |
| Permian | 81 | 3897 | 2197 | 64 | 36 | 6094 | 1.3 | 59.9 | 38.9 | 0.6 | 0.6 |
| Appalachia | 60 | 127 | 5052 | 2 | 98 | 5179 | 1.2 | 73.2 | 26.4 | 0.2 | 0.2 |
| Eagle Ford | 35 | 1344 | 1112 | 55 | 45 | 2456 | 1.4 | 65.0 | 34.0 | 0.5 | 0.5 |
| Bakken | 23 | 1361 | 444 | 75 | 25 | 1805 | 1.3 | 64.9 | 34.5 | 0.4 | 0.2 |
| Anadarko | 70 | 548 | 1237 | 31 | 69 | 1785 | 3.9 | 70.5 | 28.7 | 0.4 | 0.4 |
| Galkynysh/ Dauletabad | 83 | 0 | 1533 | 0 | 100 | 2017 | 4.1 | 74.1 | 22.6 | 0.5 | 2.8 |

### 3.7 Uncertainty contributions and breakdown of selection criteria

As described in Section 2, the total daily uncertainty $u_\Phi$ is determined from individual uncertainty components quantifying the impact of the enhancement patterns in the context of systematic biases or single measurement precision ($u_E$), absolute wind speed and wind direction knowledge or variability ($u_{v,\mathrm{abs}}$ and $u_{v,\mathrm{dir}}$), and pressure or topography ($u_{\rho_{dry}}$) on the emission estimates (see Equation 2). The mean percentage variance contribution for a given component $c$ is defined as the mean over all contributing days of the daily $(\frac{u_c}{c})^2/(\frac{u_\Phi}{\Phi})^2$. Besides the emission and production values used to determine the leakage rates, the individual mean percentage variance contributions to the emission estimates are summarised in Table 1 for the regions under consideration in this study. As can be seen, the main sources of uncertainty are given by the variable enhancement patterns related to precision and accuracy and by the spatial and temporal variability of the absolute wind speed. Due to the exclusion of days with mean wind direction changes larger than $30°$ during the considered 2 hour time window of wind history, the contribution of wind direction variability is small. The same is true for the relative impact of topography on the emission estimates at least for the regions analysed here.

Section 2 also describes the filter criteria for selecting the data in order to ensure reliable emission estimates. Most excluding are the ones that filter out days with too few data coverage over the corresponding region. To determine the subsequent order of the leftover filters, the criteria excluding the most days of the remaining data set are successively identified. The results are summarised in Figure 12 for different oil and gas plays under consideration. The filter criteria ordered by exclusionary power for all regions combined are: 1) too few data, 2) too high background scatter $\sigma(E_b)$, 3) too high or too low wind velocity $v$, 4) too large asymmetry $|\bar{E}^N_{b,p} - \bar{E}^S_{b,p}|$ with respect to the equator, 5) considerable wind direction change within the 2 hour time window of wind history, 6) too large daily uncertainty $u_\Phi$. For the individual regions, the respective filter sequences are similar with a maximum permutation of two criteria compared to the overall sequence.

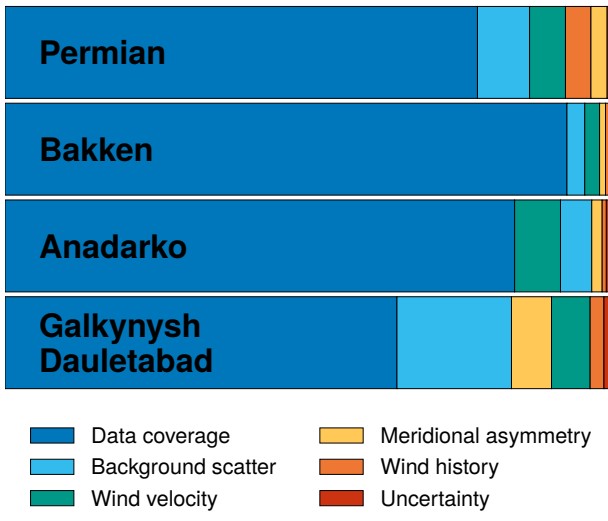

**Figure 12.** Relative contributions of the selection criteria sorted by importance for different regions. See main text for details.

## 4 Conclusions

We have analysed regional atmospheric methane enhancements over large oil and gas production areas derived from daily measurements in the shortwave infrared spectral range of the TROPOMI instrument onboard the Sentinel-5 Precursor satellite to estimate the mean emissions for the analysed regions during the period 2018/2019. The analysis benefits from TROPOMI's unique combination of high precision, accuracy, and spatiotemporal coverage allowing for the systematic detection of sufficiently large emission sources in a single satellite overpass.

To assess the climate impact of the oil and gas industry, the determined emission estimates were related to the combined oil and gas production of the considered regions in terms of energy content to infer the respective fugitive emission rates. A summary of the results is given in Table 1 and illustrated in Figure 13 showing the leakage rates for the different regions analysed in comparison to bottom-up estimates for the entire United States. In addition to regions where the inferred fugitive emission rates are reasonably consistent with the bottom-up estimates and likely below the break-even rate for immediate climate benefit (Appalachia, Permian, Bakken, and Eagle Ford), we have also identified regions that probably exceed this range (Anadarko and Galkynysh/Dauletabad) rendering a climate benefit over all time frames for these production areas questionable. The results suggest that it is possible to reduce methane emissions below the break-even leakage rate at which the climate impacts of gas/oil and coal coincide, if sufficient technological efforts are undertaken and appropriate industrial practices are employed. On the other hand, this does not seem to have been achieved in all production regions yet. In particular, relatively newly developed oil and gas plays appear to have larger leakage rates as compared to more mature production areas. As a consequence, there is still potential to reduce fugitive methane emissions from natural gas and petroleum systems worldwide.

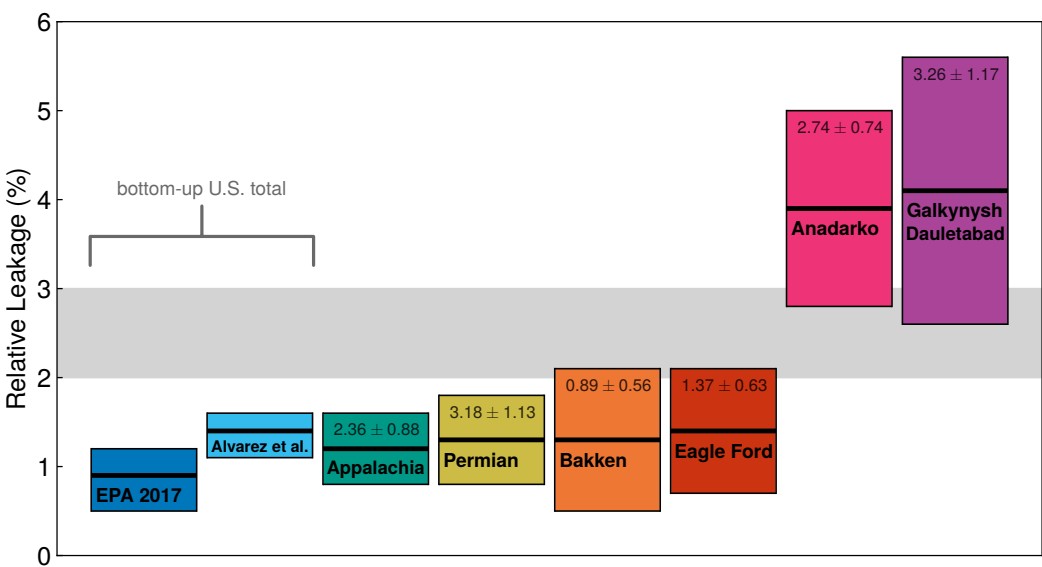

**Figure 13.** Summary of the results for the different regions analysed in this study and a comparison to bottom-up estimates for the entire United States. All leakage rates are calculated relative to combined oil and gas production in terms of energy content. The respective absolute emissions in $\text{Mt yr}^{-1}$ are shown in the upper area of the bars for the individual regions. The assumed break-even range for immediate climate benefit is shown in grey.

The self-imposed goal of large parts of the oil and gas companies to reduce the leakage rate below $1\%$ has probably not yet been achieved in the measured regions.

Due to the inherent heterogeneity of methane leakage among the energy sector depending on operating conditions and proce-dures, it is difficult to specify typical leakage rates and to reliably assess the climate footprint of the natural gas and petroleum

5   industry as a whole, which is essential for developing a sagacious environmental and energy policy. Further studies including other regions and longer time series are needed to evaluate the sustainability of the oil and gas industry unambiguously by obtaining a better sampling of the leakage distribution. Better knowledge of the relationships between leakage and production practices or basin development would also serve to improve current spatially and temporally resolved emission databases. In order to achieve these objectives, satellite measurements, ideally supplemented by frequent aircraft and ground-based measure-

10  ments, can make an important contribution. The analysis of the main sources of uncertainty of the satellite-based emission and leakage estimates suggests that future missions with improved precision and spatial resolution may have the potential to refine the current capabilities of emission monitoring from space by further reducing uncertainties. However, any emission estimation requires accurate knowledge of the wind speed and direction at an adequate horizontal and vertical resolution, which is not directly available by satellite observations and has to be provided by external sources.

*Data availability.* The methane data set presented in this publication can be accessed via http://www.iup.uni-bremen.de/carbon_ghg/products/ tropomi_wfmd/.

*Author contributions.* OS designed and operated the TROPOMI/WFMD satellite retrievals, performed the data analysis, interpreted the results, and wrote the paper. MB, MR, SV, HB, and JPB provided significant conceptual input to the design of the TROPOMI/WFMD

5  satellite retrievals, the interpretation, and the improvement of the paper. All authors discussed the results and commented on the paper.

*Competing interests.* The authors declare that they have no conflict of interest.

*Acknowledgements.* This publication contains modified Copernicus Sentinel data (2018, 2019). Sentinel-5 Precursor is an ESA mission implemented on behalf of the European Commission. The TROPOMI payload is a joint development by ESA and the Netherlands Space Office (NSO). The Sentinel-5 Precursor ground-segment development has been funded by ESA and with national contributions from The

10  Netherlands, Germany, and Belgium. The research leading to the presented results has in part been funded by the ESA projects GHG-CCI, GHG-CCI+, and Methane+, the Federal Ministry of Education and Research project AIRSPACE (grant 01LK1701B), the DLR project "S5P Datennutzung" (50EE1811A), and by the State and the University of Bremen.

We acknowledge the use of the meteorological ERA5 reanalysis from the European Centre for Medium-Range Weather Forecasts (ECMWF) and the oil and natural gas production data from the U.S. Energy Information Administration (EIA).

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
