# Peer review of "Remote sensing of methane leakage from natural gas and petroleum systems revisited"

_Atmospheric Chemistry and Physics, 2020_

## Referee Comment (RC1) · Anonymous Referee #1 · 22 May 2020

The paper by Schneising et al. uses recent years of TROPOMI satellite CH4 data to estimate fugitive emission rates for CH4 from several large oil and gas extraction fields in the US as well as one in central Asia. They employ a previously demonstrated semi-Lagrangian integral approach using wind data from ECMWF analyses, and relate the estimated CH4 emissions to the reported energy production of the fields in order to assess whether the overall emissions (fugitive plus combustion) are an improvement upon a coal-based system in terms of global warming potential. Overall, this is a very nicely done paper. The data are relatively new and unique, the analysis approach is sound and well-described, including uncertainty estimates, and the paper is exceptionally well-written. Although the climate break-even point for leaks relative to production is a somewhat qualitative metric for assessing the industry and its technology, it does

serve to put different fields and prior estimates on a common footing from which relative advantages can be compared. The paper's conclusions are robust, interesting, and probably useful for policy considerations.

In terms of comments and suggestions for revision of the paper, there is not much more to recommend. The paper was a pleasure to review. The one thing, however, that the authors should address is the findings of the recent paper by Zhang et al. (2020). I realize that the latter appeared after the submission of the Schneising et al. paper, but the two have so much in common, yet rather different messages, that it would be remiss not to have the authors discuss it. This could, perhaps, be handled in Discussions, but since many readers will just download the manuscript and forego the Discussions, they would miss out on a key aspect of the work.

To initiate the discussion, it is very interesting that the two papers, using quite different analysis methods (Zhang et al. used TROPOMI data with a Bayesian inverse), produce almost identical (within uncertainties) estimates for total fugitive $CH_4$ emissions for the Permian. This is remarkable in its own right. What is more striking, however, is the difference in messaging. The two produce very different qualitative assessments (i.e., 'spin') of their policy implications. In one case (Schneising et al.), the emissions are framed as acceptable: 'below the break-even rate for immediate climate benefit,' while in Zhang et al. the fugitive emissions are regarded with alarm: 'Permian Basin appears to be associated with insufficient infrastructure . . . leading to extensive venting and flaring (Fig. 3), which contributes to high methane emissions.' Besides the technical difference of how $CH_4$ emissions are related to the total energy production (divisor for the waste percentage), it seems that preconceptions are influencing how the results are discussed. While both papers reach the obvious conclusion that much more can and should be done to limit fugitive emissions, we need a balanced consideration of how to use the satellite data to that effect. Note how the press took off with the Zhang et al. paper: https://www.newscientist.com/article/2241347-fracking-wells-in-the-us-are-leaking-loads-of-planet-warming-methane/. At a minimum the authors Schneising et

al. should put the percentage waste estimates on the same divisor, address any other discrepancies, e.g., the relation to prior estimates, and, hopefully, initiate an informed discussion on how to use these results for progress in reducing effective emissions.

Y. Zhang, R. Gautam, S. Pandey, M. Omara, J. D. Maasakkers, P. Sadavarte, D. Lyon, H. Nesser, M. P. Sulprizio, D. J. Varon, R. Zhang, S. Houweling, D. Zavala-Araiza, R. A. Alvarez, A. Lorente, S. P. Hamburg, I. Aben, D. J. Jacob, Quantifying methane emissions from the largest oil-producing basin in the United States from space. Sci. Adv. 6, eaaz5120 (2020).
* * *

---

## Referee Comment (RC2) · Anonymous Referee #2 · 29 May 2020

The paper addresses the climate benefit of changing energy generation from coal to oil or natural gas, which has been particularly active in the United States. The authors quantify the fugitive methane emissions from the oil and natural gas industry using satellite retrievals of XCH4 from Sentinel 5P TROPOMI and systematically assess whether these fugitive emissions are below the break-even rate for direct climate benefit. The paper thus addresses important and timely questions on climate actions and presents how these can be verified with novel satellite measurements of atmospheric composition; therefore, I consider the paper to be well in the scope of ACP.

The authors have put together a concise study and used valid, clearly outlined methods for obtaining the results. All methods and calculations have been described with necessary details and thus seem to fully allow traceability of the results. While the

method used for emission estimates is not novel, the newly available TROPOMI data and, on the other hand, ongoing changes in the oil and gas production merit revisiting the topic, as has been done in this paper. The literature references are, for the most part, substantial which puts the paper well in context within the research field and gives appropriate credit to the work that this study builds upon or otherwise touches. However, I do agree with Reviewer #1 on the importance of considering the recently published Zhang et al. (2020) results in this work – this addition will further increase the quality of this paper. Regarding the Reviewer's comment on how to communicate the results, I appreciate Schneising et al.'s discreet voice of describing the results and conclusions.

The paper reads exceptionally well and has a clear structure and logic that is easy to follow. The language is fluent and precise, and delightfully versatile. The title is informative and reflects the contents of the paper sufficiently, and the essential elements of the study are summarised in the abstract. I found all figures and the table useful. Overall, I only have a few minor comments and questions on some details because I read the paper with great interest. These are listed below.

Specific comments:

Section 2: The authors do a thorough job in explaining criteria that they have found necessary for selecting the data in order to produce reliable emission estimates. I'm curious how demanding these criteria are; which one is the most excluding or is this case-dependent? I assume these criteria were set by experimenting. Are the criteria equally good for all cases?

Page 11, lines 2–5: The authors comment that the actions to reduce fugitive emissions have been successful. Are these actions potentially mentioned or described in any citable source? I think this is a strong point towards verifying climate actions and reaching company sustainability goals but would be interesting to know what kind of actions have taken place and when.

Section 3.7: Does the selected resolution of the XCH4 gridding affect your results? Did you experiment with different grid resolutions?

Technical corrections:

Figures 4, 5, 7, 8, 10 captions: coordinate E should be W

Figure 11 caption: coordinate W should be E

Page 4, line 27: straight-forward should be straightforward

Eq. (1) lacks period from the end.

Eq. (3) lacks period from the end.

Page 7, line 24: remove comma after "note"

Page 8, line 2: 5 –> five

Figure 6 (also elsewhere): spelling of the time interval could be harmonised (2009-2011 but 2018/2019).

Page 14, line 10: a –> an

---

## Short Comment (SC1) · 8 Jun 2020

Dear authors, this is a very good and interesting manuscript. The one discussion I am missing though is on the choice of background correction. The paper merely states that the background is "suitable". What makes the chosen background suitable? Where other background areas tested and if so, how do they effect the result? Particularly since the background in methane is so large compared to the excess methane above the target area, I would very much appreciate a small paragraph on this in the revised paper. Thank You!

---

## Author Comment (AC1) · 25 Jun 2020

Thank you for your interest in the details of our method. Since it is an automated processing chain, the (prototype) background region is defined in a generic way upwind of the source (blue box in Figure 2): It has the same position relative to the source for all days and all investigated regions in the transformed coordinate system (zonal direction matches wind direction) with the same meridional extent as the plume region. Whether the selected background is actually suitable to reliably estimate the emissions for a given day is automatically evaluated using certain selection criteria described in Section 2 (Page 7) of the manuscript. This has been made clearer in the revised version when the background region is introduced. We also added a paragraph in Section 3.7 discussing the exclusionary power of the filter criteria including those concerning the

background region. The impact of the zonal extent of the background region was not explicitly tested but the effect is considered small due to the implemented automatic "suitability check" on a daily basis.
* * *

---

## Author Response (AR1)

Bremen, June 25, 2020

**Letter to the Editor of paper acp-2020-274**

Dear Editor,

on behalf of all co-authors I have prepared this document, which provides the point-by-point responses to the reviews and the short comment as well as a highlighting of all changes made in the revised manuscript. The emission estimation has been extended by one month, as ERA5 for December 2019 has been published in the meantime. As a consequence, the mean estimates of some of the regions have slightly changed.

Best regards,

Oliver Schneising
(corresponding author)

**1 Final response to referee comments on paper acp-2020-274**

We would like to thank both reviewers for their efforts in thoroughly reviewing our manuscript and for their constructive comments, which helped to further improve the paper. In the following, we provide answers and clarifications to all comments of the referees (repeated in italics).

**Anonymous Referee #1**

***Reviewer:*** *In terms of comments and suggestions for revision of the paper, there is not much more to recommend. The paper was a pleasure to review. The one thing, however, that the authors should address is the findings of the recent paper by Zhang et al. (2020). I realize that the latter appeared after the submission of the Schneising et al. paper, but the two have so much in common, yet rather different messages, that it would be remiss not to have the authors discuss it. This could, perhaps, be handled in Discussions, but since many readers will just download the manuscript and forego the Discussions, they would miss out on a key aspect of the work.*

*To initiate the discussion, it is very interesting that the two papers, using quite different analysis methods (Zhang et al. used TROPOMI data with a Bayesian inverse), produce almost identical (within uncertainties) estimates for total fugitive CH4 emissions for the Permian. This is remarkable in its own right. What is more striking, however, is the difference in messaging. The two produce very different qualitative assessments (i.e., 'spin') of their policy implications. In one case (Schneising et al.), the emissions are framed as acceptable: "below the break-even rate for immediate climate benefit", while in Zhang et al. the fugitive emissions are regarded with alarm: "Permian Basin appears to be associated with insufficient infrastructure ... leading to extensive venting and flaring (Fig. 3), which contributes to high methane emissions." Besides the technical difference of how CH4 emissions are related to the total energy production (divisor for the waste percentage), it seems that preconceptions are influencing how the results are discussed. While both papers reach the obvious conclusion that much more can and should be done to limit fugitive emissions, we need a balanced consideration of how to use the satellite data to that effect. Note how the press took off with the Zhang et al. paper: `https://www.newscientist.com/article/2241347-fracking-wells-in-the-us-are-leaking-loads-of-planet-warming-methane/`. At a minimum the authors Schneising et al. should put the percentage waste estimates on the same divisor, address any other discrepancies, e.g., the relation to prior estimates, and, hopefully, initiate an informed discussion on how to use these results for progress in reducing effective emissions.*

**Authors:** We agree that the findings of the Zhang et al. (2020) paper (which appeared after the submission of our paper) should be discussed and set in relation to our results. In the revised version, we have added a paragraph describing the differences in the two approaches and comparing the corresponding results. We also explain, why we think that our method of computing the leakage rate, which we have already used in earlier work, is better suited to assess the climate impact compared to coal. The new paragraph reads as follows:

"Concurrent with our study, Zhang et al. (2020) also quantified methane emissions from the Permian basin using a different data set and an alternative inversion method combining information from the operational TROPOMI methane product and prior emission estimates within a Bayesian framework. Despite these quite distinct approaches, their total emission estimate of $2.9 \pm 0.5\,\mathrm{Mt\,yr^{-1}}$ based on satellite observations from May 2018 to March 2019 agrees within uncertainties with our estimate. If we restrict our analysis to this specific period, the consistency becomes even better and we get the almost identical estimate of $2.8\,\mathrm{Mt\,yr^{-1}}$ with our method, which is independent of prior knowledge. Therefore, the corresponding absolute results are considered very robust. However, there is a crucial difference in the calculation and subsequent interpretation of the leakage rate: while our rate (1.3%) is calculated relative to combined oil and gas production in terms of energy content (Schneising et al., 2014), the rate of Zhang et al. (2020) is larger (3.7%) and appears more alarming because it is put in relation to natural gas production only. With this alternative divisor we would also get a leakage rate of 3.7% (as can be determined from Table 1). But as the Permian is dominated by oil production, we consider the total energy approach to be better suited to assess the climate impact compared to coal in general. Otherwise, the energy content of the extracted oil would be neglected and a pure oil play (with an infinitesimal fraction of not marketed but vented natural gas) would have a leakage rate of 100%. For a pure natural gas play, however, both approaches to determine the leakage rate coincide."

**Anonymous Referee #2**

**Specific comments**

***Reviewer:*** *... I do agree with Reviewer #1 on the importance of considering the recently published Zhang et al. (2020) results in this work - this addition will further increase the quality of this paper. Regarding the Reviewer's comment on how to communicate the results, I appreciate Schneising et al.'s discreet voice of describing the results and conclusions.*

**Authors:** We have added a corresponding paragraph in the revised version. See also answers to Referee #1.

***Reviewer:*** *Section 2: The authors do a thorough job in explaining criteria that they have found necessary for selecting the data in order to produce reliable emission estimates. I'm curious how demanding these criteria are; which one is the most excluding or is this case-dependent? I assume these criteria were set by experimenting. Are the criteria equally good for all cases?*

**Authors:** We have added a dedicated paragraph and a corresponding figure to Section 3.7 of the manuscript:

"Section 2 also describes the filter criteria for selecting the data in order to ensure reliable emission estimates. Most excluding are the ones that filter out days with too few data coverage over the corresponding region. To determine the subsequent order of the leftover filters, the criteria excluding the most days of the remaining data set are successively identified. The results are summarised in Figure 12 for different oil and gas plays under consideration. The

filter criteria ordered by exclusionary power for all regions combined are: 1) too few data, 2) too high background scatter $\sigma(E_b)$, 3) too high or too low wind velocity $v$, 4) too large asymmetry $|\bar{E}_{b,p}^N - \bar{E}_{b,p}^S|$ with respect to the equator, 5) considerable wind direction change within the 2 hour time window of wind history, 6) too large daily uncertainty $u_\Phi$. For the individual regions, the respective filter sequences are similar with a maximum permutation of two criteria compared to the overall sequence."

***Reviewer:*** *Page 11, lines 2-5: The authors comment that the actions to reduce fugitive emissions have been successful. Are these actions potentially mentioned or described in any citable source? I think this is a strong point towards verifying climate actions and reaching company sustainability goals but would be interesting to know what kind of actions have taken place and when.*

**Authors:** We have added a sentence describing the kind of actions that have taken place with corresponding references to Section 3.2:

"The systematic measures implemented proactively by coalitions of oil and gas companies since 2014 to continuously reduce methane emissions include additional leak detection and repair campaigns, replacement or upgrade of high-emitting devices, and reduction of venting or flaring, to direct toward the ambitious goal of achieving a leakage rate not exceeding 1% across the natural gas supply chain (including a maximum of 0.3% from upstream operations) by 2025 (ONE Future, 2019; Oil and Gas Climate Initiative, 2019)."

***Reviewer:*** *Section 3.7: Does the selected resolution of the XCH4 gridding affect your results? Did you experiment with different grid resolutions?*

**Authors:** We have not experimented with different grid resolutions and expect only a small impact on the results. The selected grid resolution was chosen because it is close to the native resolution of the TROPOMI instrument.

**Technical corrections**

***Reviewer:*** *Figures 4, 5, 7, 8, 10 captions: coordinate E should be W*

**Authors:** Has been changed in the revised version.

***Reviewer:*** *Figure 11 caption: coordinate W should be E*

**Authors:** Has been changed.

***Reviewer:*** *Page 4, line 27: straight-forward should be straightforward*

**Authors:** Has been changed.

***Reviewer:*** *Eq. (1) lacks period from the end.*

**Authors:** Has been added.

***Reviewer:*** *Eq. (3) lacks period from the end.*

**Authors:** Has been added.

*Reviewer: Page 7, line 24: remove comma after "note"*

**Authors:** Has been removed.

*Reviewer: Page 8, line 2: 5 → five*

**Authors:** Done.

*Reviewer: Figure 6 (also elsewhere): spelling of the time interval could be harmonised (2009-2011 but 2018/2019).*

**Authors:** We would like to keep the original spelling: "/" for a list of specific years and "-" for a period (which is longer than two years) with start and end year.

*Reviewer: Page 14, line 10: a → an*

**Authors:** Done.

**Correspondence:** O. Schneising (oliver.schneising@iup.physik.uni-bremen.de)

**Abstract.** The switch from the use of coal to natural gas or oil for the energy generation potentially reduces the greenhouse gas emissions and thus the impact on global warming and climate change because of the larger energy content per $CO_2$ molecule emitted. However, the climate benefit over coal is offset by methane ($CH_4$) leakage from natural gas and petroleum systems, which reverses the climate impact mitigation if the rate of fugitive emissions exceeds the compensation point at which the
5  global warming resulting from the leakage and the benefit from the reduction of coal combustion coincide. Consequently, an accurate quantification of the $CH_4$ emissions from the oil and gas industry is essential to evaluate the suitability of natural gas and petroleum as bridging fuels on the way to a carbon-neutral future.

   We show that regional $CH_4$ release from large oil and gas fields can be monitored from space by using dense daily recurrent measurements of the TROPOspheric Monitoring Instrument (TROPOMI) onboard the Sentinel-5 Precursor satellite to quantify
10  emissions and leakage rates. The average emissions for the time period 2018/2019 from the five most productive basins in the United States, the Permian, Appalachia, Eagle Ford, Bakken, and Anadarko are estimated to be  $3.18 \pm 1.13$ $\mathrm{Mt\,yr^{-1}}$, $2.36 \pm 0.88\,\mathrm{Mt\,yr^{-1}}$,  $1.37 \pm 0.63\,\mathrm{Mt\,yr^{-1}}$, $0.89 \pm 0.56\,\mathrm{Mt\,yr^{-1}}$, and  $2.74 \pm 0.74\,\mathrm{
[revised manuscript text omitted]